# Diagnostic and Therapeutic Challenges in a Patient with Radiation Retinopathy Complicated by Corticosteroid-Induced Central Serous Chorioretinopathy

**DOI:** 10.3390/medicina58070862

**Published:** 2022-06-28

**Authors:** Michał Chrząszcz, Natalia Mackiewicz, Weronika Pociej-Marciak, Bożena Romanowska-Dixon, Agnieszka Kubicka-Trząska, Maciej Gawęcki, Izabella Karska-Basta

**Affiliations:** 1Department of Ophthalmology, Clinic of Ophthalmology and Ocular Oncology, Faculty of Medicine, Jagiellonian University Medical College, 31-551 Cracow, Poland; m.a.chrzaszcz@gmail.com (M.C.); mackiewicz.nat@gmail.com (N.M.); weronika.pociej@gmail.com (W.P.-M.); romanowskadixonbozena1@gmail.com (B.R.-D.); agnieszka.kubicka-trzaska@uj.edu.pl (A.K.-T.); 2Dobry Wzrok Ophthalmological Clinic, 80-402 Gdańsk, Poland; maciej@gawecki.com

**Keywords:** central serous chorioretinopathy, retinoblastoma, macular edema, corticosteroids, radiation retinopathy

## Abstract

Central serous chorioretinopathy (CSC) is a common chorioretinal disorder. It has been postulated that impaired retinal pigment epithelium and hyperpermeability of the choriocapillaris may be involved in the development of CSC, but the exact pathomechanism has not been established. We report an unusual case of a middle-aged man who developed CSC after triamcinolone acetonide injection for macular edema. Edema developed as a late complication of radiation retinopathy after brachytherapy for childhood retinoblastoma. Steroid treatment is an important risk factor for CSC, but the underlying causative mechanisms have not been fully elucidated. It is important to increase the awareness of this link among clinicians who prescribe exogenous corticosteroids, irrespective of the route of administration.

## 1. Introduction

Central serous chorioretinopathy (CSC) is a common chorioretinal disorder that belongs to the spectrum of pachychoroid diseases [1]. It primarily involves the accumulation of fluid in the subretinal space caused by a focal leakage at the level of the retinal pigment epithelium (RPE). This leads to reduced visual acuity, color vision disturbances, and metamorphopsia. The exact pathomechanism of CSC has not been fully elucidated so far [2,3,4], but some well-known risk factors include male sex, psychological stress with Type A personality, smoking, pregnancy, and cardiovascular and gastrointestinal diseases [5,6,7].

Interestingly, several previous studies reported a relationship between CSC and treatment with exogenous and endogenous corticosteroids administered via different routes, including oral, intravenous, intramuscular, epidural, intraarticular, topical dermal, inhaled, ophthalmic, and after dexamethasone intravitreal implant [5,6,7,8]. A precise molecular mechanism underlying the steroid-induced onset of CSC is currently unknown. However, corticosteroids are known to inhibit collagen formation, a key component of the Bruch’s membrane. They may also cause RPE dysfunction by altering the transepithelial resistance to water and ions [9]. Disruption in the regulation of choroidal blood flow is also possible as corticosteroids may alter the production of nitric oxide, prostaglandins, and free radicals [10].

To the best of our knowledge, this is the first case report of CSC following intravitreal triamcinolone acetonide (TA) injection for radiation macular edema in a patient treated for retinoblastoma in childhood. Our case illustrates the challenges of CSC diagnosis in the only remaining eye with extensive fundus abnormalities caused by radiation retinopathy, with cataract as a complication of the previous treatment, and with difficult access to the fundus.

## 2. Case Report

A 43-year-old man with a childhood history of bilateral retinoblastoma (Figure 1), enucleation of the right eye, and two sessions of brachytherapy with cobalt-60 in the left eye (with the subsequent five sessions of xenon photocoagulation) presented with symptoms of blurred vision. His best corrected visual acuity (BCVA) was 0.2 by Snellen chart, and the intraocular pressure was normal. A slit-lamp examination of the anterior segment revealed early corticonuclear cataract. Posterior-segment examination revealed macular edema with radiation retinopathy and multiple chorioretinal scars after laser photocoagulation. There were no signs of inflammation in the anterior segment of the left eye. Fundus examination by indirect ophthalmoscopy revealed no soft exudates or retinal hemorrhages (Figure 2A). Optical coherent tomography (OCT) showed macular edema in the right eye (Figure 2B and Figure 3). Fluorescein angiography report described a hyperfluorescent area on the temporal side of the fovea of the left eye, but there were no nonperfusion areas or signs of vasculitis. The patient had no history of hypertension or diabetes and did not use any medications that could induce macular edema. Therefore, we assumed that the macular edema was caused by previous radiation therapy. Based on literature data, we used anti-vascular endothelial growth factor (VEGF) agents as the first-line approach [11,12]. Moreover, considering that our patient had only one eye, we decided on anti-VEGF therapy to avoid complications such as secondary cataract or glaucoma. It was reported that up to four injections of an anti-VEGF agent at intervals of 1 month improve macular edema and BCVA in patient with radiation maculopathy [13]. Therefore, we administered three intravitreal injections of bevacizumab (1.25 mg/0.05 mL). However, no functional and clinical improvement was achieved. Therefore, the anti-VEGF therapy was switched to intravitreal TA injections (4 mg/0.1 mL). During the four-year follow-up, the patient received eight TA intravitreal injections. No increase in intraocular pressure was observed after any of the intravitreal TA injections. Each injection led to transient functional improvement. OCT showed a significant reduction in central retinal thickness and resolution of macular edema (Figure 4). BCVA improved to 0.6. During that time, the patient underwent a phacoemulsification of cataract with intraocular lens implantation and yttrium-aluminum-garnet (YAG) laser posterior capsulotomy due to posterior capsule opacification. 

Two months after the last TA injection, the patient presented to our center with sudden vision deterioration (metamorphopsia, watery vision in the lower field of view, particularly during reading). BCVA was maintained at 0.5. OCT revealed serous neurosensory retinal detachment in the upper part of the macula, without the involvement of the fovea centralis (Figure 5). The choroid is thicker in the upper part of the macula where the subretinal fluid is present than in the subfovea and the lower part of the macula (Figure 5). Fundus autofluorescence showed an area of hypoautofluorescence in chorioretinal scars induced by brachytherapy as well as an area of hyperautofluorescence in the superior temporal vascular arcades due to the presence of subretinal fluid (Figure 6). Early phase fluorescein angiography revealed a focal hyperfluorescence with the inkblot pattern of dye leakage above the macula without the involvement of the foveal center (Figure 7). Fundus autofluorescence, fluorescein angiography, and OCT findings confirmed the diagnosis of CSC. A direct focal 200 µm spot of 150 mW and 0.1 s duration laser photocoagulation at the leakage point was performed. Additionally, considering that anti-VEGF agents modify the permeability properties of the vascular choroid and that they have a well-known efficacy in CSC, even in the absence of choroidal neovascularization, we performed bevacizumab injection. This resulted in the total resolution of the serous retinal detachment (Figure 8). BCVA improved to 0.6.

## 3. Discussion

Patients with CSC are more likely to have been previously exposed to systemic or local corticosteroid treatment. Various routes of local steroid administration have been described, including inhaled, intranasal, epidural, intra-articular, topical dermal, and periocular [5]. Recently, a case of acute recurrent CSC that developed after treatment with corticosteroid enemas and suppositories was reported [14]. Moreover, there have been two case reports of CSC after intravitreal steroid administration: one of them described CSC induced by vitrectomy and TA injection and the other presented a patient with an exacerbation of CSC after intravitreal TA injection [15,16]. It is possible that TA injected into the vitreous cavity affects the RPE through the neural retina, as previously reported by an in vitro study [17]. Following the introduction of dexamethasone intravitreal implant as a novel treatment option, several cases of CSC caused by this intervention have been reported [8]. A multicenter Japanese survey compared CSC in patients treated and patients not treated with steroids [18]. The study included 538 eyes in 477 patients with CSC followed for 3 months or longer. Patients treated with steroids were older than those not receiving steroids (mean age, 56 years and 53 years, respectively). Moreover, the male prevalence was lower in steroid-treated patients (39 men among the total number of 74 participants vs. 305 men among the total number of the 403 participants not taking steroids). The affected eyes of steroid-treated patients also had multiple pigment epithelial detachments. Approximately 41% of the steroid group presented with bilateral CSC, as compared with only 7.4% of the other group. The authors concluded that steroids could cause more severe CSC by affecting choroidal vessels and impairing the RPE [18]. Steroid-induced CSC can induce pigment epithelial detachment and increase choroidal thickness [19].

As mentioned above, a molecular mechanism underlying the link between steroid use and CSC onset has not been fully elucidated. O’Brien et al. [9] postulated corticosteroid-induced inhibition of collagen formation, possibly leading to outer blood-retinal barrier breakdown. The barrier function of the RPE (another component of the outer blood-retina barrier) may be also disturbed by ion and water transport that can be impaired by corticosteroids [20]. This may lead to subretinal fluid accumulation as well as serous RPE detachment that is typical for CSC. The most popular hypothesis nowadays is that the mechanism underlying the development of CSC starts with nonspecific disturbance in the choroidal circulation. Corticosteroids have been postulated to increase fibroblastic growth, resulting in capillary fragility in the choroidal vessels and suboptimal choriocapillaris function [21]. Another possible explanation is the modulation of the choroidal circulation. It is known that choroidal blood flow is regulated by the sympathetic and parasympathetic systems. Steroids, on the other hand, are known to act synergistically with the sympathetic system and antagonistically with the parasympathetic system, causing alterations in the production of the vascular modulator, free radicals, and nitric oxide synthase [10]. These interactions may lead to choroidal vascular spasms, changing vessel permeability and/or perfusion [22]. Finally, Zhao et al. [23] reported that an intravitreal corticosterone injection into rat eyes induced choroidal enlargement. The authors hypothesized that the underlying mechanism is the glucocorticoid-induced activation of the mineralocorticoid receptor pathway, which leads to the upregulation of the calcium-dependent endothelial vasodilatory potassium channel KCa2.3 and, consequently, to choroidal vasodilation [23].

The available clinical experience suggests that steroid treatment should be administered with caution not only in patients with retinal pigment epithelial detachment and increased choroidal thickness [18]. In our case, clinical examination and OCT were challenging due to radiation cataract, insufficient pupillary response to mydriatics, brachytherapy complications, and scarred chorioretinal lesions after photocoagulation for retinoblastoma, but fluorescein angiography proved to be crucial in establishing the correct diagnosis.

## 4. Conclusions

Steroid treatment constitutes an easily modifiable risk factor for CSC. Although the diagnosis of CSC in the eye with concomitant chorioretinal disorders is challenging, it should always be considered in corticosteroid-treated patients, irrespective of the route of drug administration.

## Figures and Tables

**Figure 1 medicina-58-00862-f001:**
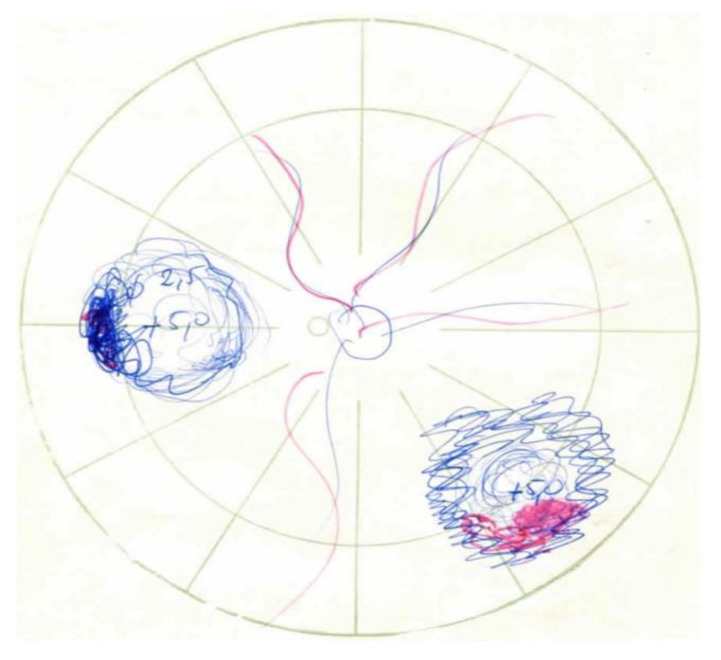
Schematic presentation of the left eye fundus in 1980, showing two foci of retinoblastoma at diagnosis. One tumor with surface bleeding was located in the inferotemporal quadrant, and the other in the nasal quadrant close to the equator.

**Figure 2 medicina-58-00862-f002:**
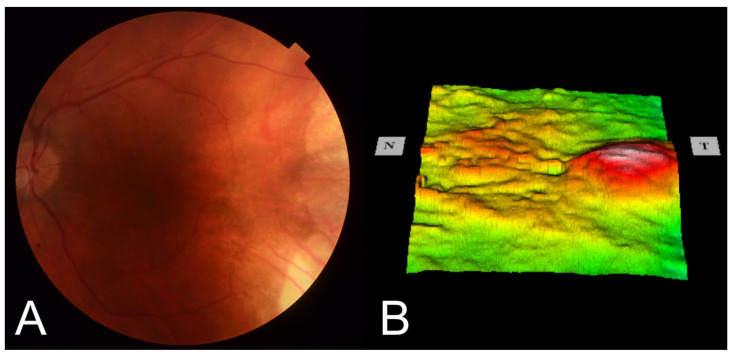
Color fundus photography (**A**) of the left eye showing chorioretinal atrophy induced by brachytherapy in the temporal part of the fundus; macular thickness map (**B**) showing macular edema (particularly in the temporal part of the macula).

**Figure 3 medicina-58-00862-f003:**
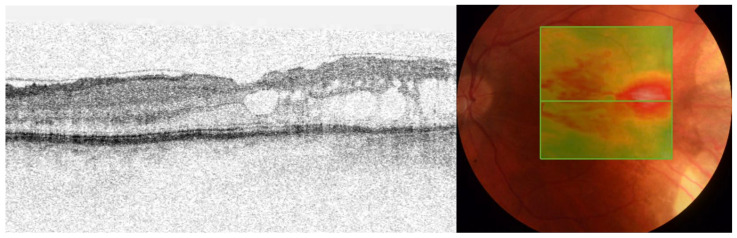
A horizontal optical coherence tomography (OCT) scan of the left eye showing cystoid macular edema with subretinal fluid, particularly in the temporal macula.

**Figure 4 medicina-58-00862-f004:**
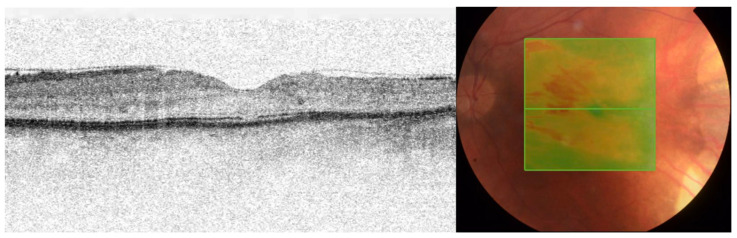
A horizontal optical coherence tomography (OCT) scan of the left eye after triamcinolone acetonide (TA) injections. Improvement in macular morphological changes and visible foveal contour with central macular edema resolution.

**Figure 5 medicina-58-00862-f005:**
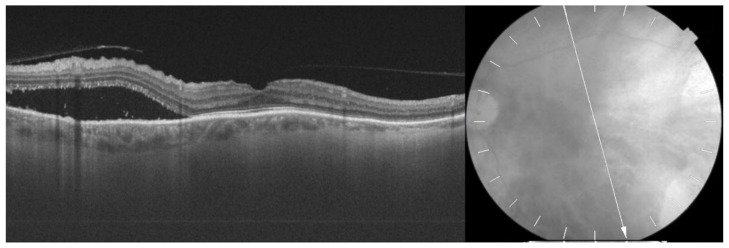
An optical coherence tomography (OCT) scan of the left eye after the last TA injection. The presence of subretinal fluid resulted in serous neurosensory retinal detachment in the upper part of the macula without the involvement of the fovea centralis.

**Figure 6 medicina-58-00862-f006:**
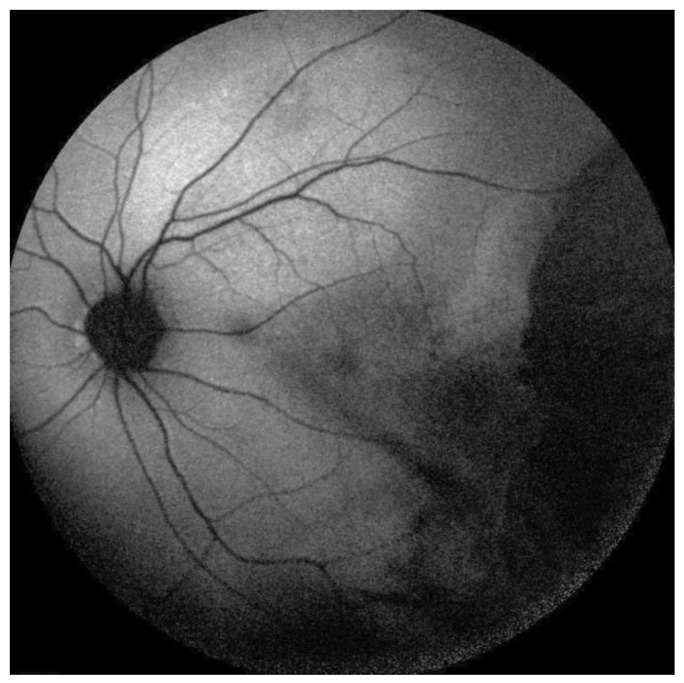
Fundus autofluorescence of the left eye showing the area of hypoautofluorescence in chorioretinal scars induced by brachytherapy as well as the area of hyperautofluorescence in the superior temporal vascular arcades due to subretinal fluid.

**Figure 7 medicina-58-00862-f007:**
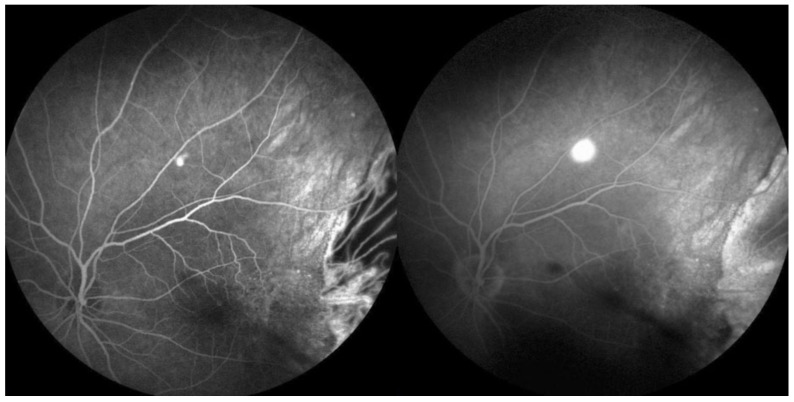
Fluorescein angiography (FA) of the left eye: early and late phases. A focal hyperfluorescence point in the early phase above the macula near the superior temporal vascular arcade with an inkblot pattern of dye leakage.

**Figure 8 medicina-58-00862-f008:**
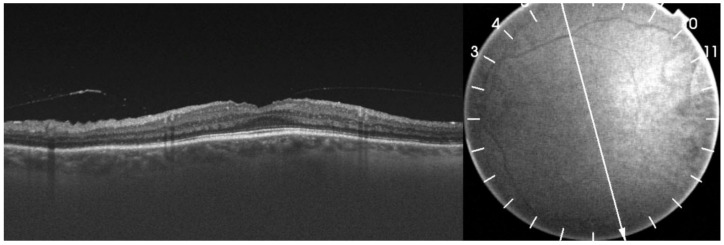
An OCT scan of the left eye after focal laser photocoagulation of the leakage point and bevacizumab injection. Note the complete resolution of subretinal fluid.

## Data Availability

Not applicable.

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
