# Peer review of "Diagnostic and Therapeutic Challenges in a Patient with Radiation Retinopathy Complicated by Corticosteroid-Induced Central Serous Chorioretinopathy"

_medicina, 2022, doi:10.3390/medicina58070862_

Round 1

Reviewer 1 Report

  1. Introduction (Page 1, Line 41): Regarding the supposition that it is the first case reported, the authors should describe the literature searching methodology they used to come to this conclusion.
  2. Case Report (Page 2, Line 53): If FA or OCT angiography was performed, please also describe its findings.
  3. Case Report (Page 2, Lines 53,54): Please provide evidence that ME was caused by radiation retinopathy.
  4. Case Report (Page 2, Lines 56,57): Please describe the reason for the intravitreal injections of bevacizumab despite no confirmation of choroidal neovascularization. In addition, please describe why bevacizumab was injected three times even though macular edema did not resolve after one dose.
  5. Case Report (Page 2, Lines 59,60): Please describe whether there was any increase in intraocular pressure during the 4-year follow-up (eight TA intravitreal injections).
  6. Case Report (Page 3, Figure 2): Please describe in which direction the OCT B-scan image is.
  7. Case Report (Page 3, Line 86): Please describe the reason why bevacizumab injection was performed at the same time as the laser treatment.
  8. Case Report (Page 4, Figure 5): I consider that FAF abbreviation is not required.
  9. Case Report (Page 4, Figure 6): I consider that FA abbreviation is not required.
  10. Discussion (Page 5, Line 126): Is the choroidal thickness increased in this case?

Author Response

1. Introduction (Page 1, Line 41): Regarding the supposition that it is the first case reported, the authors should describe the literature searching methodology they used to come to this conclusion. 

Thank you for this comment. We performed a comprehensive search of the PubMed database, including both controlled vocabulary (MeSH) and keyword terms such as “macular edema” AND “radiation retinopathy” AND “triamcinolone” AND “central serous chorioretinopathy”.

2. Case Report (Page 2, Line 53): If FA or OCT angiography was performed, please also describe its findings.

Thank you for this suggestion. We added information on the examinations performed and how we reached the diagnosis (lines 56-60). Figure 2 was also added, presenting chorioretinal atrophy on color fundus photography and macular edema on macular thickness map. Only an FA report was available due to the change of the equipment and the lack of the results themselves.

3. Case Report (Page 2, Lines 53,54): Please provide evidence that ME was caused by radiation retinopathy.

Thank you for this suggestion. We added relevant information in the “Case report” section of the manuscript (lines 55-62) .

4. Case Report (Page 2, Lines 56,57): Please describe the reason for the intravitreal injections of bevacizumab despite no confirmation of choroidal neovascularization. In addition, please describe why bevacizumab was injected three times even though macular edema did not resolve after one dose.

Thank you for this comment. There are different types of treatment for radiation maculopathy, including intravitreal anti–vascular endothelial growth factor (anti-VEGF) agents, sub-Tenon or intravitreal injection of triamcinolone acetonide, intravitreal dexamethasone implantation, or photodynamic therapy using verteporfin. Based on literature data, we used anti-VEGF agents as the first-line approach. Moreover, considering that our patient had only one eye with the original meniscus, we decided on anti-VEGF therapy to avoid complications such as secondary cataract or glaucoma.

Mashayekhi et al. reported that anti-VEGF injections every month (up to 4 injections) improve macular edema and BCVA. Such management resulted in a decrease of macular edema in 56% of patients. Based on these reports, we decided to administer three injections in our patient. Relevant information was added in the manuscript (lines 64-68).

Murray, T.G.; Latiff, A.; Villegas, V.M.; Gold, A.S. Aflibercept for radiation maculopathy study: a prospective, randomized clinical study. Ophthalmol Retina 2019, 3, 561–566. doi: 10.1016/j.oret.2019.02.009.

Fallico, M.; Chronopoulos, A.; Schutz, J.S.; Reibaldi, M. Treatment of radiation maculopathy and radiation-induced macular edema: A systematic review. Surv Ophthalmol 2021, 6, 441–460. doi: 10.1016/j.survophthal.2020.08.007.

Mashayekhi, A.; Rojanaporn, D.; Al-Dahmash, S.; Shields, C.L.; Shields, J.A. Monthly intravitreal bevacizumab for macular edema after iodine-125 plaque radiotherapy of uveal melanoma. Eur J Ophthalmol 2014, 24, 228–234. doi: 10.5301/ejo.5000352.

5. Case Report (Page 2, Lines 59,60): Please describe whether there was any increase in intraocular pressure during the 4-year follow-up (eight TA intravitreal injections).

No increase in intraocular pressure was observed after any of the intravitreal TA injections. The information was added in line 72-73.

6. Case Report (Page 3, Figure 2): Please describe in which direction the OCT B-scan image is.

Thank you for this valuable comment. This was a horizontal OCT scan. Relevant information was added in Figures 3 and 4, line 89-96.

7. Case Report (Page 3, Line 86): Please describe the reason why bevacizumab injection was performed at the same time as the laser treatment.

Thank you for bringing this to our attention. Bevacizumab injection was performed at the same time as laser treatment because the patient had only one eye and OCT angiography was not technically feasible. We thus wanted to avoid the risk of vision deterioration if laser photocoagulation proved ineffective. Because CSC is believed to originate from the choroidal vasculature, intravitreal anti-VEGF injections have been suggested by Torres-Soriano et al. as possible treatment for CSC by modifying choroidal vascular permeability. There have been reports showing the efficacy of anti-VEGF therapy in CSC, even in the absence of CNV (van Rijssen et all). We added relevant information in lines 111-113.

Mitzy E. Torres-Soriano, Gerardo García-Aguirre, Verónica Kon-Jara, Orlando Ustariz-Gonzáles, Maura Abraham-Marín, Michael D. Ober & Hugo Quiroz-Mercado A pilot study of intravitreal bevacizumab for the treatment of central serous chorioretinopathy (Case reports) Graefe's Archive for Clinical and Experimental Ophthalmology volume 246, pages1235–1239 (2008)

Van Rijssen, T.J.; van Dijk, E.H.C.; Yzer, S.; Ohno-Matsui, K.; Keunen, J.E.E.; Schlingemann, R.O.; Sivaprasad, S.; Querques, G.; Downes, S.M.; Fauser, S.; et al. Central serous chorioretinopathy: Towards an evidence-based treatment guideline. Prog. Retin. Eye Res. 2019, 100770. 

8. Case Report (Page 4, Figure 5): I consider that FAF abbreviation is not required.

Thank you, the abbreviation was deleted as per your comment.

9. Case Report (Page 4, Figure 6): I consider that FA abbreviation is not required.

Thank you, the abbreviation was deleted as per your comment.

10. Discussion (Page 5, Line 126): Is the choroidal thickness increased in this case?

Thank you for this valuable comment. Indeed, subfoveal choroidal thickness was not increased in our patient. We believe this is due to previous brachytherapy. According to literature data, brachytherapy may result in choroidal thinning. Lee et al reported choroidal thinning 6 months after ruthenium brachytherapy in patients treated for choroidal melanoma. However, in our patient, in the upper part of the macula where subretinal fluid is present, the choroid is thicker than in the subfovea or the lower part of the macula, as shown in Figure 5. Relevant information was added in lines 101-103.

Ji Hwan Lee, Sung Chul Lee, Arthur Cho, Ki Chang Keum, Yang-Gun Suh, Christopher Seungkyu Lee Association Between Choroidal Thickness and Metabolic Activity on Positron Emission Tomography in Eyes With Choroidal Melanoma, Am J Ophthalmol. 2015 Dec;160(6):1111-1115.e2. doi: 10.1016/j.ajo.2015.08.031.

Reviewer 2 Report

The authors describe an interesting case of CSCR following an intravitreal injection of Triamcinolone.  The manuscript is well documented and nicely written. CSCR following steroid injection is not uncommon and can occur in any disease where an intravitreal steroid is indicated. 

Few comments:

Page 3 line 76: "...the patient presented to our center with sud- 76 den vision deterioration and metamorphopsia."Kindly mention the visual acuity when the patient presented with CSCR. As the neurosensory detachment is not involving the foveal center, it is unlikely to have a drop in central vision.

Page 3 line 86: What was the rationale for treating with bevacizumab? As the leak is neither sub-foveal nor does FFA show any neovascularisation.

Author Response

1. Page 3 line 76: "...the patient presented to our center with sudden vision deterioration and metamorphopsia."Kindly mention the visual acuity when the patient presented with CSCR. As the neurosensory detachment is not involving the foveal center, it is unlikely to have a drop in central vision.

Thank you for this comment. Sudden vision deterioration was evidenced by metamorphopsia and watery vision in the lower field of view (particularly during reading). BCVA was 0.5. The information was added in lines 98-99.

2. Page 3 line 86: What was the rationale for treating with bevacizumab? As the leak is neither sub-foveal nor does FFA show any neovascularisation.

Thank you for bringing this to our attention. Bevacizumab injection was performed at the same time as laser treatment because the patient had only one eye and OCT angiography was not technically feasible. We thus wanted to avoid the risk of vision deterioration if laser photocoagulation proved ineffective. Because CSC is believed to originate from the choroidal vasculature, intravitreal anti-VEGF injections have been suggested by Torres-Soriano et al. as possible treatment for CSC by modifying choroidal vascular permeability. There have been reports showing the efficacy of anti-VEGF therapy in CSC, even in the absence of CNV (Ref Van Rijssen). We added relevant information in lines 111-113

Mitzy E. Torres-Soriano, Gerardo García-Aguirre, Verónica Kon-Jara, Orlando Ustariz-Gonzáles, Maura Abraham-Marín, Michael D. Ober & Hugo Quiroz-Mercado A pilot study of intravitreal bevacizumab for the treatment of central serous chorioretinopathy (Case reports) Graefe's Archive for Clinical and Experimental Ophthalmology volume 246, pages1235–1239 (2008)

Van Rijssen, T.J.; van Dijk, E.H.C.; Yzer, S.; Ohno-Matsui, K.; Keunen, J.E.E.; Schlingemann, R.O.; Sivaprasad, S.; Querques, G.; Downes, S.M.; Fauser, S.; et al. Central serous chorioretinopathy: Towards an evidence-based treatment guideline. Prog. Retin. Eye Res. 2019, 100770. 

Reviewer 3 Report

This is a clinical case reporting a CSC after intravitreal triamcinolone. This is not a very usual finding in CSC and this is an interesting point that could be explored in more detail - why systemic CCT use causes more CSC than local CCT use?

Author Response

Thank you for this valuable comment. We added a new paragraph in the discussion section describing the current concepts and hypotheses on the mechanisms underlying corticosteroid-induced development of CSC (see lines 155-176).

Although the link between the use of systemic and local corticosteroids (both endogenous and exogenous) and CSC onset has been well documented, it remains unknown why CSC is more common with systemic than with local corticosteroid use. On the one hand, this may be explained by the fact that with local use (topical or intranasal corticosteroids), only a small amount of the drug enters the circulation and reaches the eye. On the other hand, it has been suggested that CSC induced by steroids may be associated with an idiosyncratic response of very susceptible individuals rather than a dose-dependent effect (Kaye et al). Cases have been reported where even very small doses of corticosteroids induced episodes of CSC, while patients on long-term treatment with high doses of oral corticosteroid may never develop CSC (van Dijk et al). This issue requires further studies.

van Dijk, E.H.C., Soonawala, D., Rooth, V., Hoyng, C.B., Meijer, O.C., de Vries, A.P.J., Boon, C.J.F., 2017c. Spectrum of retinal abnormalities in renal transplant patients using chronic low-dose steroids. Graefes Arch. Clin. Exp. Ophthalmol. 255 (12), 2443–2449

Rebecca Kaye, Shruti Chandra, Jay Sheth, Camiel J.F. Boon, Sobha Sivaprasad, Andrew Lotery, Central serous chorioretinopathy: An update on risk factors,pathophysiology and imaging modalities, Progress in Retinal and Eye Research, 79, November 2020, 100865, https://doi.org/10.1016/j.preteyeres.2020.100865

Reviewer 4 Report

The authors reported a case of a middle age man, who developed CSC after TA injection. The paper is well written in patient's ocular history and ocular examination. However, I would suggest to the authors to discuss more about the potential mechanisms and clinical relevant of the finding to improve the ''Interest to the readers''. In general, I think it is a good case report.

Author Response

Thank you for this comment. The potential mechanisms and clinical relevance of our case report were discussed thoroughly at the end of the discussion section (see lines: 156-176). This was supported by relevant literature, as listed below:

O'Brien, P.; Young, R.C.; Ghafoori, S.D.; Harper, C.A.; Wong, R.W. Central serous retinopathy associated with topical oral corticosteroid use: a case report. J Med Case Rep 2019, 13, 201. doi: 10.1186/s13256-019-2143-3.

Bouzas, E.A.; Karadimas, P.; Pournaras, C.J. Central serous chorioretinopathy and glucocorticoids. Surv Ophthalmol 2002, 47, 431–448. doi: 10.1016/S0039-6257(02)00338-7.

Sanjay, S.; Gowda, P.B.; Rao, B.; Mutalik, D.; Mahendradas, P.; Kawali, A.; Shetty, R. "Old wine in a new bottle" - post COVID-19 infection, central serous chorioretinopathy and the steroids. J Ophthalmic Inflamm Infect 2021, 11, 14. doi: 10.1186/s12348-021-00244-4.

Nakatsuka, A.S.; Khanamiri, H.N.; Lam, Q.N.; El-Annan, J. Intranasal Corticosteroids and central serous chorioretinopathy: a report and review of the literature. Hawaii J Med Public Health 2019, 78, 151–154.

Carvalho-Recchia, C.A.; Yannuzzi, L.A.; Negrão, S.; Spaide, R.F.; Freund, K.B.; Rodriguez-Coleman, H.; Lenharo, M.; Iida, T. Corticosteroids and central serous chorioretinopathy. Ophthalmology. 2002, 109, 1834–1837. doi: 10.1016/s0161-6420(02)01117-x.

Zhao, M.; Célérier, I.; Bousquet, E.; Jeanny, J.C.; Jonet, L.; Savoldelli, M.; Offret, O.; Curan, A.; Farman, N.; Jaisser, F.; Behar-Cohen, F. Mineralocorticoid receptor is involved in rat and human ocular chorioretinopathy. J Clin Invest 2012, 122, 2672–2679. doi: 10.1172/JCI61427.

Round 2

Reviewer 1 Report

Dear Authors,

Thank you for addressing my comments and suggestions.

Author Response

We would like to thank you for your comments, which have enhanced the content value of the manuscript. We are glad that we met your expectations.